# Influence of Structure and Composition of Woven Fabrics on the Conductivity of Flexography Printed Electronics

**DOI:** 10.3390/polym13183165

**Published:** 2021-09-18

**Authors:** Ana María Rodes-Carbonell, Josué Ferri, Eduardo Garcia-Breijo, Ignacio Montava, Eva Bou-Belda

**Affiliations:** 1Textile Research Institut, Universitat Politècnica de València, 46022 Valencia, Spain; 2Instituto Tecnológico del Textil (AITEX), 03801 Alcoy, Spain; josue.ferri@aitex.es; 3Instituto Interuniversitario de Investigación de Reconocimiento Molecular y Desarrollo Tecnológico (IDM), Universitat Politècnica de València, 46022 Valencia, Spain; egarciab@eln.upv.es; 4Department of Textile and Paper Engineering, Universitat Politècnica de València, Plaza Ferrándiz y Carbonell s/n., 03801 Alcoy, Spain; imontava@txp.upv.es (I.M.); evbobel@upvnet.upv.es (E.B.-B.)

**Keywords:** flexography, e-textiles, wearables, printed-electronics, textiles, electronic textiles

## Abstract

The work is framed within Printed Electronics, an emerging technology for the manufacture of electronic products. Among the different printing methods, the roll-to-roll flexography technique is used because it allows continuous manufacturing and high productivity at low cost. Nevertheless, the incorporation of the flexography printing technique in the textile field is still very recent due to technical barriers such as the porosity of the surface, the durability and the ability to withstand washing. By using the flexography printing technique and conductive inks, different printings were performed onto woven fabrics. Specifically, the study is focused on investigating the influence of the structure of the woven fabric with different weave construction, interlacing coefficient, yarn number and fabric density on the conductivity of the printing. In the same way, the influence of the weft composition was studied by a comparison of different materials (cotton, polyester, and wool). Optical, SEM, color fastness to wash, color measurement using reflection spectrophotometer and multi-meter analyses concluded that woven fabrics have a lower conductivity due to the ink expansion through the inner part of the textile. Regarding weft composition, cotton performs worse due to the moisture absorption capacity of cellulosic fiber. A solution for improving conductivity on printed electronic textiles would be pre-treatment of the surface substrates by applying different chemical compounds that increase the adhesion of the ink, avoiding its absorption.

## 1. Introduction

Printed electronics (PE) refers to the technology that allows electronic device manufacturing through a printing process. PE is one of the fastest growing technologies in the world as it provides different printing techniques for manufacturing low-cost and large-area flexible electronic devices [1]. In recent years, flexible electronics technology has attracted considerable attention since it can be applied to wearable devices including flexible displays, flexible batteries and flexible sensors [2,3] in different areas such as aerospace and automotive, biomedical, robotics, and health applications [4]. Among them, wearable electronic textiles (e-textiles) are of great significance since they provide better comfortability, durability and lighter weight as well as maintaining desirable electrical properties [5].

The PE printing technique choice must be done according to the electronic application (e.g., small, thin, lightweight, flexible, and disposable, etc.), the manufacturing cost and volume. Additionally, the main materials (inks/pastes and substrates) must meet certain requirements, depending on the printing technology selected and the final application.

PE technologies are divided into contact techniques (e.g., flexography, gravure printing and soft lithography techniques), where the printing plate is in direct contact with the substrate and non-contact techniques (e.g., screen printing, aerosol printing, inkjet printing, laser direct writing), where only the ink contacts the substrate [6]. Those techniques suitable for roll-to-roll (R2R) processing, such as flexography, are especially attractive since they offer continuous manufacturing and high productivity [7].

Flexographic printing is known for depositing a wide range of thicknesses with the same resolution. The impression cylinder, plate cylinder, anilox roller, doctor blade and inking unit are the main parts of the flexographic printing process [1], as illustrated in Figure 1. Variables associated with the flexographic printing process include print speed, print force/engagement, anilox cell volume, anilox force/engagement as well as the ink and substrate properties [8]. Those variables have a direct impact on the print’s morphological and electrical behavior. The print uniformity has a considerable influence on the final functionality of the device [9]. Within the context of printing electronics onto fabrics, it must be highlighted the challenge of durability and withstanding bending, stretching, abrasion and washing [10].

Numerous reviews and books were previously published, considering printed electronics on substrates such as glass, metal, paper or polymers [4,7]. Different studies were focused on the application of printed electronics onto textiles to obtain electrical devices such as capacitive sensors [11], perovskite solar cells [12] or RFID tags [13], but by using other printing techniques, such as ink jet or screen printing which are more common.

However, the incorporation of the flexography printing technique for printed electronics in the textile field is still very recent and there are not enough studies for its application. Continuing our previous work [14], this research is focused on the influence of the textile substrate parameters on the electrical performance of the printing.

The most used textile type for garments and even for technical applications is woven fabrics, where two sets of perpendicular yarns are crossed and interweave each other to create a coherent and stable structure [15]. In the context of this study, the term ‘structure’ refers to binding patterns of interlacing threads in woven fabrics, also considering internal structural features of the threads involved. Therefore, the structure of the woven fabric is determined, among other factors, by the weave construction, the interlacing coefficient, the density of threads in the fabric and the characteristics of warp and weft threads [16]. All the other physical (crimp, thickness, physical density, mass of unit area, porosity, etc.), mechanical (breaking strength and elongation, tensity, forces of rupture, resistance to abrasion, etc.) and permeability properties (permeabilities of gasses, liquids, light, sound, energy, water vapor, bacteria, etc.) are affected by previously mentioned selection [17]. Fabric texture and composition affect the porosity and strongly influence the textile characteristics such as the fabric mass, thickness, draping ability, stress–strain behavior, or air permeability. The surface topography of fabrics is responsible for their functionality—appearance and handle, wettability, soiling behavior and cleanability, abrasion resistance and wear. Topographical characteristics of the fabrics strongly depend on their construction parameters such as the type and fineness of filaments, yarn fineness, yarn density, and the type of weave. These characteristics have strong influence on, and in many cases, control the wetting properties [18].

It is important that printing ink has good adhesion to the substrate [19]. This becomes particularly complex in printing electronics onto fabrics where the intrinsic porous structures and texture characteristic of textiles affects the diffusion and penetration of conductive ink, being able to deteriorate the printing precision and electrical performance of conductive lines [20].

Related to the fabric structure, it was proved in other printing techniques, such as screen-printing, that the smallest pore size and roughness shows a higher printing precision and lower electrical resistance of printed conductive lines [20].

According to material composition, natural fibers such as cotton tend to absorb the ink more readily than synthetic fibers such as polyester, due to hydrophilic and wettability properties of cellulosic fibers [21]. Generally, synthetic textile materials have smooth, tight surfaces that offer little texture for ink adhesion [19].

However, there are not systematic investigations about the relationship between the electrical performance of a flexo printed textile with the structure of the woven fabric (i.e., interlacing coefficient, yarn count and weft density) as well as with the material composition of the weft.

With the aim of establishing this relation, different textiles were specifically manufactured varying their structural parameters and materials. After that, a controlled flexographic printing process was performed by using a silver electrical ink. Finally, fabrics were physically and electrically examined and compared.

The research and the obtained results are presented from two different approaches. On the one hand, an approach is followed from a perspective focused on distinguishing the impact of the structural parameters of the woven fabric on the conductivity of the printing. Structural parameters include, but are not limited to, weave construction, interlacing coefficient, yarn count and fabric density. On the other hand, from a point of view based on establishing the influence of the weft material composition on the conductivity, a comparison between fabrics with different weft materials (cotton, polyester, and wool) was also performed.

Physical and electrical analyses were carried out for both approaches. Several methodologies were used including optical, FE-SEM, color fastness to wash, and color measurement using reflection spectrophotometer or multi-meter analyses. Significant results were obtained and therefore studied.

It is believed that these findings will provide some important support for printing electronic devices on woven textiles by using the flexographic technique.

## 2. Materials and Methods

### 2.1. Materials

With the aim of establishing the influence of structure and composition of woven fabrics on the conductivity of flexographic printed electronics, eleven textile types were specifically manufactured for the research (Textiles Joyper S.L., Cocentaina, Spain), to be used as substrates for the printing.

On the one hand, eight different textiles structures were defined by varying the following constructional parameters: weave construction, weft yarn count, and weft density. On the other hand, textiles alternatives of the material used for the weft were added into the composition of three fabrics while maintaining the fabric structure.

Table 1 shows the main characteristics of these fabrics, where test samples from T1–T8 correspond to textile structure variations and textiles from T9–T11 to alternative weft materials. It should be noted that polyester yarn was used in the warp in all samples.

The weave construction of textile materials is the regular structure produced by a pattern (unit cell) of interlaced threads repeating at regular intervals in two transversal directions [22]. The weave interlacing coefficient, KL, which depends on weave construction, is calculated by Equation (1):(1)$$\mathrm{KL}=\frac{i}{w_{1}\times w_{2}}$$
where *i* is the number of interlacing points in the weave repeat, *w*_1_ is the number of ends in the weave repeat, and *w*_2_ is the number of picks in the weave repeat [23]. The research was focused on the two most traditional and commonly used weaves—plain and twill. The plain weave is the basic weave where one warp yarn is lifted over one weft yarn. The interlacing is opposite in all neighboring cells. Plain weave allows the highest possible number of interlacing. The twill weave has a pattern of diagonal lines; each warp yarn lifts over more than one weft, so the diagonal lines in fabric reach high densities [24].

Table 2 completes the main characteristics of the fabrics, adding the characteristics of the textile yarns in a detailed way for a better understanding.

Yarn number or yarn count refers to the thickness of a yarn and it is determined by its mass per unit length [25]. The fabric samples used in this study were produced by variation of the weft yarn count (167 dtex/333 dtex), without changing the warp count (167 dtex).

The density of the warp and weft is defined by the number of warps ends per cm and the number of picks per cm.

Therefore, the structure variations among samples from T1–T8 are the following: the weave construction, as samples from T1–T4 are plain and T5–T8 are twill; the yarn count used in for the weft, which is thicker in samples T4, T5, T7 and T8; and the weft density, which is lower in samples T1, T3, T5 and T7.

Meanwhile, while maintaining the structure parameters, the difference among samples T9–T11 is the weft composition, which is made from polyester in T9, from cotton in T10 and from wool in T11.

For the research, other structural parameters have been accordingly obtained to complete the characterization of the woven fabric samples. Cover factor (the degree of fabric fullness) is the proportion of the fabric area covered by warp and weft yarns. It means that in practice, cover factor is calculated independently for warp and weft yarn by the proportion of fabric area covered by those yarn, according to Equation (2):Cf = Cfwa + Cfwe − Cfwa × Cfw (2)
as Cfwa = Dwa × dwa and Cfwe = Dwe × dwe, where Dwa and Dwe are densities of warp and weft and dwa and dwe are the diameter of warp and weft yarns, respectively. The cover factor directly depends on the yarn density and the yarn count [26]. Finally, the fabric weight was obtained according to the standard ISO 3801 by measuring the textile mass per unit area.

Regarding the ink, flexographic printing technology requires low-viscosity printing inks, which allows regular ink flow in the printing unit. Viscosity is generally lower than 0.05–0.5 Pa·s [27]. Same aqueous flexo-printable conductive ink, PFI-RSA6012—silver ink from Novacentrix (Austin, United States), was used in all prints to ensure comparable results. Details can be consulted in Table 3. The selection was made considering high conductivities and stretching properties for printing electronics on flexible substrates. It should be highlighted that there is a low supply on the market due to the novelty of the application of printed electronics in the textile through the flexography technique as stated above.

The ink contains silver nanoparticles and was formulated for high conductivity, fast curing, and improved levelling at lower printing speed.

### 2.2. Flexographic Electronic Printing

The manufacturing technology that is used is based on the flexographic printing technique of thick film. Flexography is a roll-to-roll direct printing technology, where an anilox roller, covered with micro-cavities on its surface, allows the collection of ink, and then is transferred to the printing plate cylinder. The specification of the anilox determines the volume of ink transferred to the printing plate. The ink is taken into these cells and the excess ink is subsequently removed by a doctor blade assembly.

For the research, one-layer flexographic prints were performed by using a printing experimental plant (K Printing Proofer, RK Print Coat Instruments Ltd., Litlington, United Kingdom). The equipment allows high quality proofs using flexography, among other printing techniques, with variable printing speeds of up to 40 m/min. Printing plates for use with the experimental plant are electronically engraved in exactly the same way as production cylinders. Using the flexo head, ink is transferred from the printing plate to a plain stereo roller and then onto the substrate. Adjustments can be made by micrometers. As developing a specific electronic device is out of the scope of this research, a plain design has been used for the test; nevertheless the printing plate and the printing roller in the industrial machine could be customized.

The experimental phase of this research consisted of the flexo-printing of the woven fabrics shown in Table 1 and Table 2 using the silver ink described in Table 3. With the objective of studying the influence of the textile substrates on the conductivity of the printing, the equipment settings were kept fixed so that they did not interfere with the results. The specific setup conditions are shown in Table 4. Printed layers were dry cured in a FED-115 air oven from BINDER at 140 °C for one minute in order to use the same curing characteristics for all the samples. Moreover, previous to the printing, a thermic treatment was applied to all the fabrics to avoid variations of size due to the curing temperature of inks. The thermic treatment applied consisted of introducing the fabrics in the same oven at 130 °C for 15 min.

### 2.3. Characterization

Once dried, the printed textile samples were physically and electrically analyzed by carrying out several studies.

Regarding physical characterization, the following measurement methods were used: optical, scanning electron microscopy (FE-SEM) (Oxford Instruments plc, Abingdon, United Kingdom), and color fastness to wash and color measurement using reflection spectrophotometer.

The optical macroscopic images were taken with a LEICA MZ APO stereomicroscope. It was used to analyze the print uniformity of each layer of the electronic printed samples.

High-resolution topographic images by SE (secondary electrons) and maps of crystalline and textural orientations by EBSD (electron backscatter diffraction) (Oxford Instruments plc, Abingdon, United Kingdom) were taken with a ZEISS ULTRA 55 Scanning Electron Microscope Field Emission Gun (field emission scanning electron microscopy (FE-SEM))) (Oxford Instruments plc, Abingdon, UK). They were used to analyze the ink penetration and the adhesion in each substrate.

Color fastness to domestic and commercial laundering was evaluated with a Gyrowash according to the standard method EN ISO 105-C06:2010. The test conditions were: temperature of 40 °C, 10 steel balls and standardized ECE soap reference without optical or chemical whitener. After the test, the printed woven textiles were dried in a forced-air circulation dryer and treated samples were compared with untreated samples visually using a grey scale, according to ISO 105-A02 standard.

Color measurement was evaluated by the determination of CIELAB coordinates according to the standard method ISO 105-J01:1997. The apparatus used was DATACOLOR DC 650 (400–700 nm) (Datacolor, New Jersey, United States) with the following conditions: illuminant D_65_/10°, diffuse measuring geometry and 6.6 mm of observation area.

With respect to electrical characterization, a usual two-terminal sensing unit was firstly considered to measure the conductivity behavior. Nevertheless, a two-wire system does not provide correct output due to variation in ambient temperature, as the resistance of the lead wires (both sides) changes unpredictably. Meanwhile, 4-wire Kelvin resistance measurement makes it possible to accurately measure resistance values less than 0.1 Ω while eliminating the inherent resistance of the lead wires connecting the measurement instrument to the component being measured [28]. For that reason, 4-wire system measurements were made. Resistance measurements were made with a FLUKE 8845A multimeter from FLUKE CORPORATION (Everett, WA, USA).

## 3. Results and Discussion

### 3.1. Electrical Characterization

For the purpose of determining the electrical conductivity of the printings an approach based on measuring the electrical resistance was considered. A low resistivity indicates a material that readily allows electric current. Table 5 shows a summary of the printing results for all the fabrics in terms of electrical resistivity. Four different samples of each fabric were measured. The orientation of the measurements on the woven fabrics were separately considered. In addition, results are graphically shown in Figure 2.

Regarding the influence of the textile structure on the electrical behavior shown in Figure 2a, both measures on warp and weft directions agree that the woven fabric that presents the higher electrical resistance, and therefore worse electrical conductivity, is the substrate coded as T8. T8 was characterized by a lower interlacing coefficient and the highest mass per unit area.

These results coincide with the conclusions about the surface properties of woven fabric on electrical performance through a screen-printed technique. The fabric substrate with the smallest pore size and roughness shows a higher printing precision and lower electrical resistance of screen-printed conductive lines [20]. Therefore, the surface structure of the fabric substrate determines to some degree, not only the printing precision of conductive lines, but its electrical properties as well.

In order to address these challenges, surface pre-treatment onto rough and porous substrates or coatings and lamination processes should be performed in order to produce a continuous conductive pathway onto the textiles [5,10].

In order to deeply explore the relation found between woven fabric density and conductivity, results were plotted, as shown in Figure 3.

According to results shown in Figure 3, with regard to the influence of the textile structure on the electrical behavior, it could be expected that the higher the mass per unit, the lower conductivity due to the ink expansion through the inner part of the textile.

Moreover, Figure 2b allows the analysis in terms of the impact of the textile composition on the electrical behavior of electronic printing through the flexographic technique. Again, resistance measures in both directions agree and show that the woven fabric T10 presents higher electrical resistance. Even though the measure on the warp direction shows better electrical conductivity, it is worse than the other textile samples made from other materials. The weft of T10 was made from cotton. This makes sense, because of the moisture absorption capacity of cellulosic fiber. In this way it can be demonstrated that cotton performs worse as the weft material for printing electronics with the flexo technique.

An alternative solution for improving conductivity on printed electronic textiles with similar problems would be pre-treatment of the surface substrates by applying different chemical compounds that increase the adhesion of the ink avoiding its absorption.

### 3.2. Physical Characterization

Focusing on analyzing the print uniformity of each layer of the electronic flexo printed woven textiles, optical macroscopic images were taken with 20 magnifications. In parallel, in order to discuss the ink penetration and the adhesion in each substrate, high-resolution topographic images were taken with Scanning Electron Microscope Field Emission Gun (FE-SEM).

Table 6 shows the images obtained for the Optical and FE-SEM characterization of the woven substrates with different textile structures. The FE-SEM (×100) images show a visual characterization of the fabric and ink, and the FE-SEM (×150) images on the right show maps of crystalline and textural orientations by EBSD for a determination of the position of the silver particles.

Regarding optical results, images in Table 6 show a total color uniformity on the substrates’ surface not depending on the textile structure. The silver shade obtained is typical from conductive inks which contains silver nanoparticles. In addition, the images reveal the porosity of the woven fabrics as they show their characteristic textiles holes, being different for each structure. It should be highlighted the picture of the substrate T8 as it is the one with more holes between the yarns forming the woven fabric, in comparison with the other samples.

In respect of FE-SEM results, thanks to images of the cross section of the substrates shown in Table 6, it can be observed how weft threads with different titles and densities are intertwined with the warp threads. These pictures reveal the enclaves formed by the interlacing of the threads in both directions, obtaining a roughness of the fabric and presenting different heights. Depending on the structure and yarn title, the enclave formed is different and it directly affects to the fabric thickness. As already stated on the fabrics’ characterization in Table 2, the FE-SEM (×100) images in Table 6 confirm the thickness is greater for fabrics whose ligament is sarge (substrates T5–T8).

Furthermore FE-SEM (×150) results with more magnification allow to observe the ink penetration and distribution in the inner part of the textile. Images demonstrate that there are particles of the silver ink between the fibers of the weft threads.

Meanwhile, Table 7 shows the images obtained for the Optical and FE-SEM characterization of the woven substrates with different materials on the weft.

Regarding ink uniformity, all the woven fabrics present a good visual level at the print in spite of the porous surface. To ensure this uniformity in printing electronics with low viscosity inks onto a rough and porous textile surface is a great challenge, due to the orientation of fibers or yarns and the change of fiber morphology constantly [5].

With respect to ink penetration, the images in Table 7 show that ink does not remain at the surface, but penetrates until the lower side of the textile. Thus, results coincide with the relation pointed out in Figure 3 about the influence of the mass per unit area on the electrical behavior. Whereas conventional printing that uses thickeners to increase the viscosity of the paste achieve deposition only on the surface, the low ink viscosity used in flexo printing penetrates to the inner of the textile, the interior of the textile. Thus, the topography of the sample plays an important role in achieving the homogeneity and continuity of the ink. Therefore, it is demonstrated that the higher the mass per unit, the lower conductivity due to the ink expansion through the inner part of the textile.

In terms of the impact of the material of the weft, the images shown in Table 7 do not show significant differences among them.

The color loss and staining resulting from desorption and/or abrasive action of the samples was evaluated according to EN ISO 105-C06:2010 tests for color fastness [29]. The grade of color fastness to domestic and commercial laundering is presented in Table 8 and Table 9.

For assessing the change in color, woven treated textiles were compared with a grey scale complying with ISO 105-A02. Regarding assessing staining, the grey scale used follows ISO 105-A03. Both scales consist of five pair of non-glossy grey color chips which illustrate the perceived color differences corresponding to fastness rating 5, 4, 3, 2 and 1. A range of 5 is given only when there is no perceived difference between the tested specimen and the original material.

According to results shown in Table 8, obtained values for change in color and staining are between 4 (Very Good) and 5 (Excellent). Therefore, it can be concluded that structural variations have not a relevant influence on the color fastness of the flexo electronic printings.

However, results shown in Table 9 demonstrate that color fastness decreases with natural fibers such as cotton and wool. The reason can be that ink adhesion improves in in synthetic fibers such as polyester thanks to the curing time after the flexography printing using 150 °C, as this type of fiber is a thermoplastic material and being its glass transition temperature around 70 °C [30].

In order to address these challenges in woven textiles with natural fibers, surface pre-treatment should be done in order to improve the ink adhesion and therefore the electrical behavior [5]. It has been proved that special pretreatment processes on the fabric substrates improve the wash fastness for other printing techniques such as digital printing [31] or ink-jet [32]. According to a previous review [33], an increment of ink volume improves the ink coverage, upgrading in this case the conductivity, nevertheless it enhances the ink wash-out effect. For this reason, the ink volume transferred to the substrate should be optimized when conductivity and color fastness to washing are the objectives. In addition, coating and lamination processes could be done to ensure the continuous conductive pathway on textiles [10].

Spectrophotometric methods are adequate and objective for determining the color values of the fabric surface [34]. To assess variations found on the printings, measurements have been performed according to ISO 105-J01:1997 General principles for measurement of surface color [35].

The CIELAB, or CIE L* a* b*, color system represents quantitative relationship of colors on three axes: L*value indicates lightness, and a* and b* are chromaticity coordinates. It is the most widely used method for measuring and ordering object color. The results of the analysis of color of the printings onto the woven fabrics are shown in Table 10. For a better understanding, results have been plotted and can be consulted at the Figure 4. On the color space diagram, L* is represented on a vertical axis with values from 0 (black) to 100 (white). The a* value indicates red-green component of a color, where +a* (positive) and −a* (negative) indicate red and green values, respectively. The yellow and blue components are represented on the b* axis as +b* (positive) and −b* (negative) values, respectively. At the centre of the plane is neutral or achromatic [36].

Lower L* indicates that the sample becomes darker, and all printed samples show low light (L*) values near to 50. It is observed that there is no correlation between the conductivity and the tone obtained from the coordinates a* and b* in contrast to silver coatings of polymeric substrates, where it is demonstrated that s shifting in the a* and b* during sintering means best conductivity [37]. On the other hand, it is observed that the sample T8/T9, the one with more weight and less conductivity, shows the lowest L* value, being darker than the other samples. This fact corroborates firstly, that variations on the fabric density have an influence on the color values [34], and secondly that the penetration of the silver particles into the textile, when the silver particles remain on the surface, thanks to emulsion resins, present values of L* between 60–70 [38].

## 4. Conclusions

It was proved that a direct relationship exists between the electrical performance of a flexographic printed textile with the structure of the woven as well as with the material composition of the weft.

After performing controlled flexographic printing processes onto woven fabrics by using a silver electrical ink, physical and electrical analysis led to the following conclusions. On the one hand, from a perspective focused on the structural parameters, the weave construction, interlacing coefficient, yarn count and fabric density, there was an effect due to the higher mass per unit of the fabric and lower conductivity due to the ink expansion through the inner part of the textile. On the other hand, based on the weft material composition (cotton, polyester, and wool), it was shown that cotton performs worse as weft material for printing electronics with the flexo technique due to the moisture absorption capacity of cellulosic fiber.

It is believed that these findings will provide some important support for printing electronic devices on woven textiles by using the flexographic technique.

The study’s next steps will consist of improving the conductivity of flexographic printed textile by carrying out surface pre-treatment, coating and lamination processes that increase the ink adhesion and therefore the electrical behavior.

## 5. Patents

FERRI, J.; RODES-CARBONELL, A.M. and MORENO, J. (2020). Dispositivo NFC flexible para la medición, almacenamiento y transmisión de datos (Spain U202130440) http://www.oepm.es/pdf/ES/0000/000/01/26/48/ES-1264864_U.pdf (accessed on 12 April 2021).

## Figures and Tables

**Figure 1 polymers-13-03165-f001:**
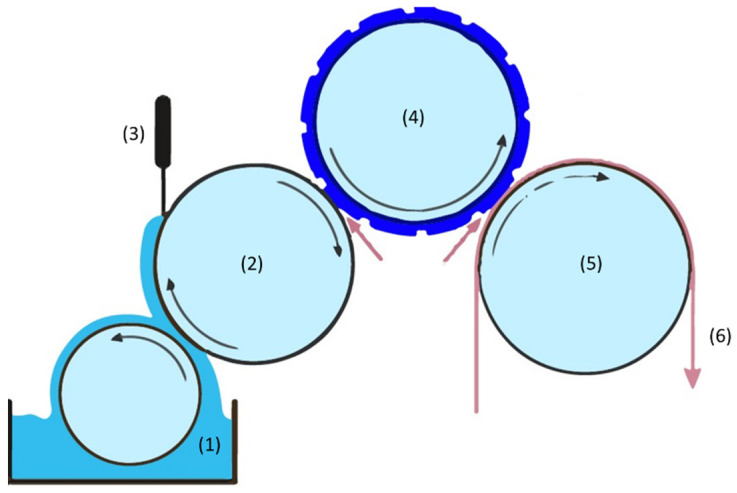
Significant parts of the flexography process: inking unit (1); anilox roller (2); doctor blade (3); plate cylinder (4); impression cylinder (5); printing substrate (6).

**Figure 2 polymers-13-03165-f002:**
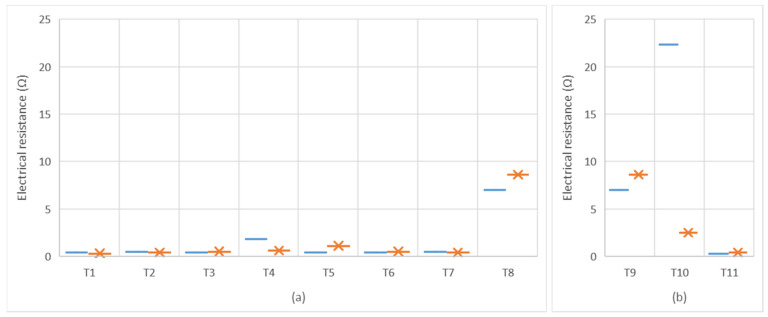
Graphic representation of the electrical resistance of the printings (
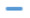
) measured on warp direction and (
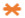
)measured on weft direction. (**a**) Electrical resistance affected by the structural variations of the textiles; and (**b**) Electrical resistance affected by the material composition of the textiles.

**Figure 3 polymers-13-03165-f003:**
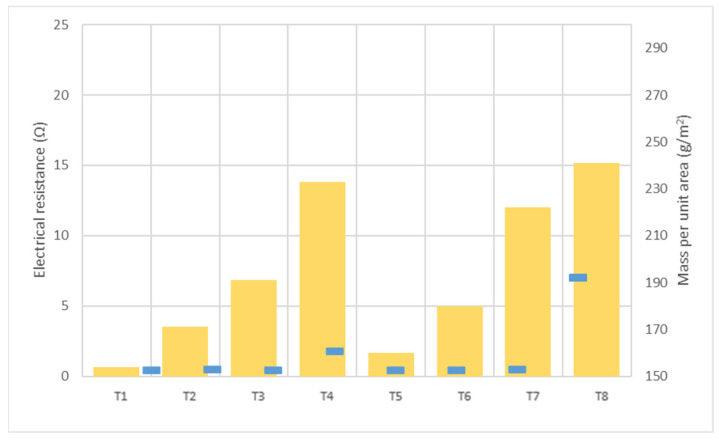
Graphic representation of the relation between electrical resistance of the printings measured on warp direction (
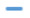
) and the mass per unit area (
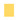
).

**Figure 4 polymers-13-03165-f004:**
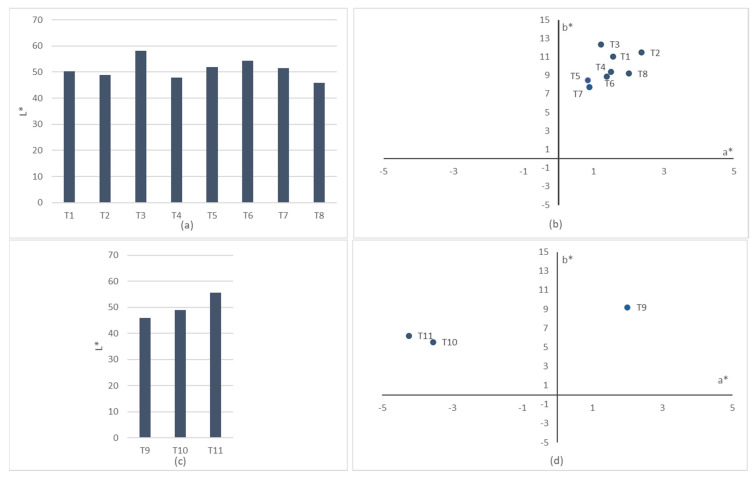
Two-dimensional CIELAB (+L*, +a *, +b*) coordinates representation. (**a**) CIELAB lightness (+L*) of textile structure variations T1–T8; (**b**) CIELAB a* and b* coordinates of textile structure variations T1–T8; (**c**) CIELAB lightness (+L*) of textile material variations T9–T11; (**d**) CIELAB a* and b* coordinates of textile material variations T9–T11.

**Table 1 polymers-13-03165-t001:** Textile substrate’s characterization (I): composition and ligament.

Substrate Code	3D Modeling	Weft Composition	Weave Construction	Graphic Representation	Interlacing Coefficient (KL)
T1	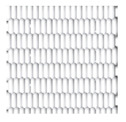	Polyester	Plain	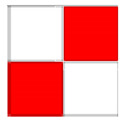	1
T2		Polyester	Plain	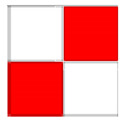	1
T3	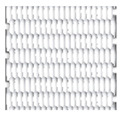	Polyester	Plain	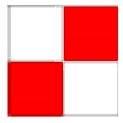	1
T4	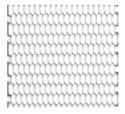	Polyester	Plain	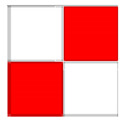	1
T5	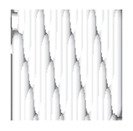	Polyester	Twill	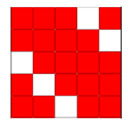	0.4
T6	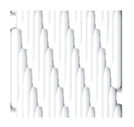	Polyester	Twill	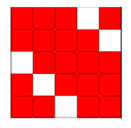	0.4
T7	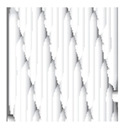	Polyester	Twill	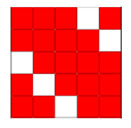	0.4
T8	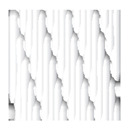	Polyester	Twill	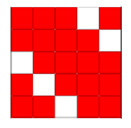	0.4
T9	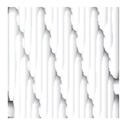	Polyester	Twill	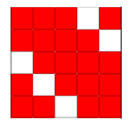	0.4
T10	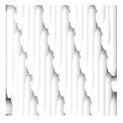	Cotton	Twill	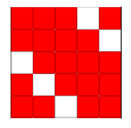	0.4
T11	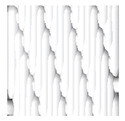	Wool	Twill	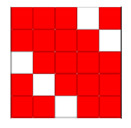	0.4

**Table 2 polymers-13-03165-t002:** Textile substrate’s characterization (II): size and weight characteristics.

Substrate Code	Warp Density (ends/cm)	Weft Density (picks/cm)	Weft Count (dtex)	Cover Factor (%)	Fabric Weight ^1^ (g/m^2^)	Thickness (µm)
T1	58.0	10.5	333.3	84.8	154	515
T2	58.8	15.3	333.3	93.6	171	550
T3	58.4	10.7	666.7	92.9	191	622
T4	57.2	15.8	666.7	103.4	233	650
T5	59.2	10.6	333.3	64.5	160	705
T6	55.9	16.0	333.3	68.3	180	725
T7	57.3	11.6	666.7	70.0	222	744
T8	56.3	16.5	666.7	77.6	241	805
T9	59.4	15.9	666.7	79.3	235	800
T10	58.8	16.2	666.7	78.5	230	795
T11	60.4	16.8	666.7	82.1	225	790

^1^ Mass per unit area determined according to the standard ISO 3801.

**Table 3 polymers-13-03165-t003:** PFI-RSA6012—Silver ink characteristics.

Ink Code	Density (g/mL)	Solids (%)	Viscosity (Pas)	Volume Resistivity (μΩ·cm)	Curing	Properties
PFI-RSA6012—Silver ink	2.22	60 (±2)	0.05–0.15 @1000 s^−1^	8–10	10–60 s 140 °C	Flexible Compatible with Polyester

**Table 4 polymers-13-03165-t004:** Printing parameters.

Ink	Anilox Volume	Resolution	Printed Area	Speed	Curing
PFI-RSA6012—Silver ink	11 cm^3^/m^2^	150 LPI	150 × 95 mm^2^	12 m/min	60 s 140 °C

**Table 5 polymers-13-03165-t005:** Electrical resistance of the printings. Measurements have been made both in warp and weft directions.

Substrate Code	Resistance (Ω) Warp Direction	Resistance (Ω) Weft Direction
T1	0.4	0.3
T2	0.5	0.4
T3	0.4	0.5
T4	1.8	0.6
T5	0.4	1.1
T6	0.4	0.5
T7	0.5	0.4
T8	7	8.6
T9	7	8.6
T10	22.3	2.5
T11	0.3	0.4

**Table 6 polymers-13-03165-t006:** Optical and FE-SEM characterization (I).

Substrate Code	Optical (×20)	FE-SEM (×100)	FE-SEM (×150)
T1	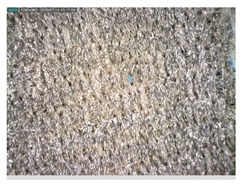	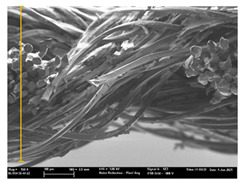	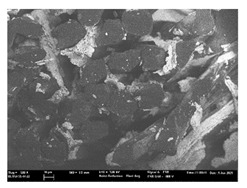
T2	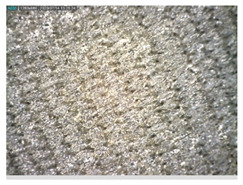	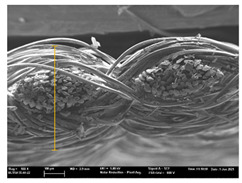	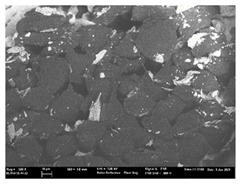
T3	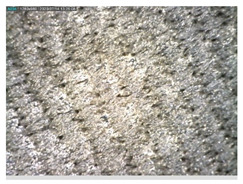	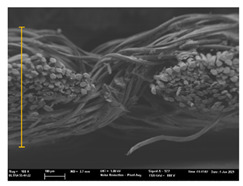	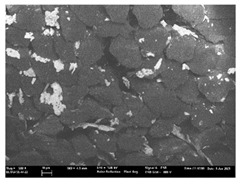
T4	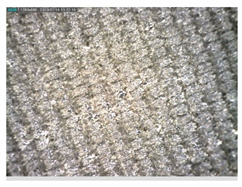	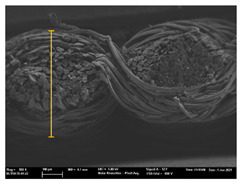	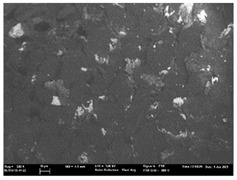
T5	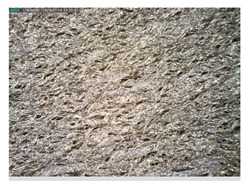	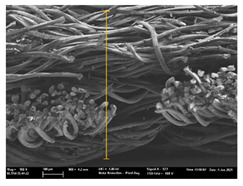	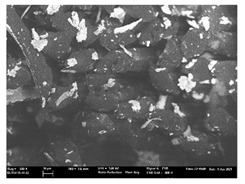
T6	* 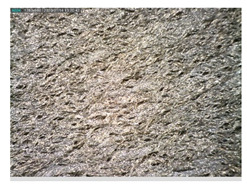 *	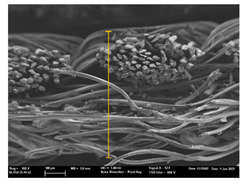	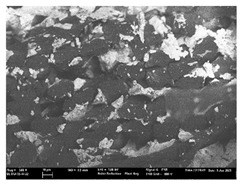
T7	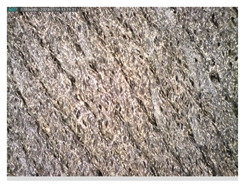	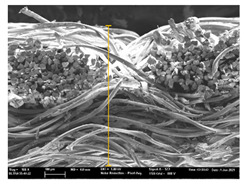	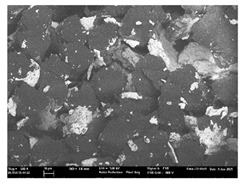
T8	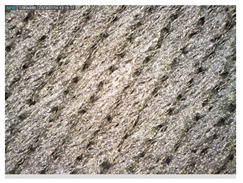	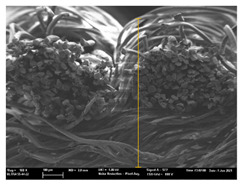	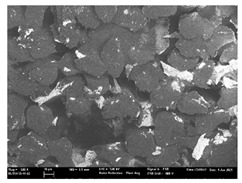

**Table 7 polymers-13-03165-t007:** Optical and FE-SEM characterization (II).

Substrate Code	Optical (×20)	FE-SEM (×100)	FE-SEM (×150)
T9	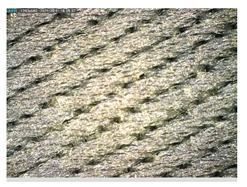	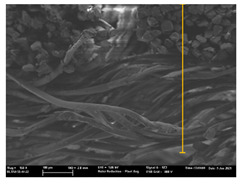	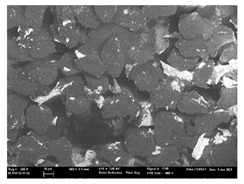
T10	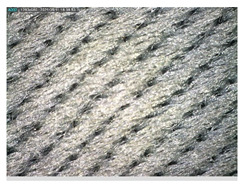	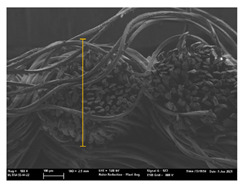	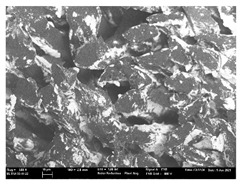
T11	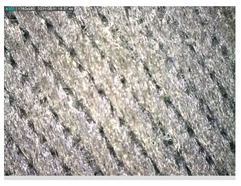	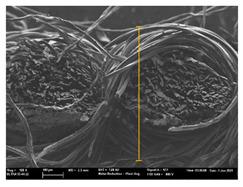	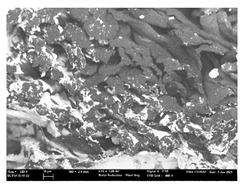

**Table 8 polymers-13-03165-t008:** Color fastness results (I).

Substrate Code	Change in Color			Staining			
		Wool	Acrylic	Polyester	Polyamide	Cotton	Acetate
T1	4–5	4–5	4–5	4–5	4–5	4–5	4–5
T2	4–5	4–5	4–5	4–5	4–5	4–5	4–5
T3	4–5	4–5	4–5	4–5	4–5	4–5	4–5
T4	4	4–5	4–5	4–5	4–5	4–5	4–5
T5	4–5	4–5	4–5	4–5	4–5	4–5	4–5
T6	4–5	4–5	4–5	4–5	4–5	4–5	4–5
T7	4–5	4–5	4–5	4–5	4–5	4–5	4–5
T8	4	4–5	4–5	4–5	4–5	4–5	4–5

**Table 9 polymers-13-03165-t009:** Color fastness results (II).

Substrate Code	Change in Color			Staining			
		Wool	Acrylic	Polyester	Polyamide	Cotton	Acetate
T9	4	4–5	4–5	4–5	4–5	4–5	4–5
T10	1	4	4	4	4	4	4
T11	1	4	4	4	4	4	4

**Table 10 polymers-13-03165-t010:** CIELAB* coordinates (D_65_/10°).

Substrate Code	L*	a*	b*
T1	50.19	1.56	11.04
T2	48.86	2.36	11.48
T3	58.07	1.21	12.36
T4	47.78	1.49	9.36
T5	51.88	0.83	8.49
T6	54.29	1.37	8.86
T7	51.58	0.87	7.71
T8	45.95	2	9.19
T9	45.95	2	9.19
T10	48.91	−3.55	5.53
T11	55.58	−4.24	6.19

## Data Availability

The data presented in this study are available on request from the corresponding author.

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
