# Peer review of "Influence of Structure and Composition of Woven Fabrics on the Conductivity of Flexography Printed Electronics"

_polymers, 2021, doi:10.3390/polym13183165_

Round 1

Reviewer 1 Report

Overall, it is an interesting study which extended understanding in the influence of structure and composition of woven fabrics on the conductivity of flexography printed electronics. However, the following concerns shall be resolved before publication.

Major:

  • A clear description for the difference among T1-T8 must be provided;
  • Line 348-350, it was indicated that “Depending on the structure and yarn title, the enclave formed is different and it directly affects to the fabric thickness. It can be observed that the thickness is greater for fabrics whose ligament is sarge (substrates T5-T8)”. How would the author draw such conclusion from Table 6? It is difficult to compare the fabric thickness based on the images listed in Table 6;
  • Line 351-352, it was indicated that “Furthermore FE-SEM results with more magnification allow to observe the ink pen-etration and distribution in the inner part of the textile”. Again, how would the author have such conclusion from Table 6, and a detail explanation must be provided. Which part of the image represent the ink particles;
  • Lines 393-397, why would the colour fastness relate to the ink adhesion? Please explain with supporting reference(s);
  • For the electrical characterization, how many specimen of each group were tested, as there is no error bars or standard deviation displayed?
  • The English of this manuscript should be improved;

Minor:

  • Table 3, the unit of volume resistivity should be µΩ·cm;
  • Please double check that the manuscript is correctly formatted. For example, should or should not skip a line between paragraphs?
  • Cf = Cfwa + Cfwe − Cfwa × Cfw (2)

Reviewer 2 Report

The paper "Influence of structure and composition of woven fabrics on the 2 conductivity of flexography printed electronics." reported about flexography printing technique in terms of various materials such as cotton, polyester, and wool. However, the reviewer thinks this paper needs more some experiments for supporting the authors’ opinion. This manuscript can be published in the journal "Polymers" after minor revision.

  1. English writing should be modified. I would suggest the author's review again to correct for awkward phrasing and language. And, please double check the typos.
  2. For helping reader’s understanding, the author explain about the proposed process in more detail.
  3. The reviewer think that it is necessary to insert exact scale bar in SEM images in tables.
  4. The author should integrate some simple electronic device with their process (e.g., LED).
  5. Printed electronics have been used in various electronics fields. Therefore the reviewer thinks the authors should talk about that in Introduction part with reference. It can improve this paper’s impact about the proposed processes. The reviewer recommends some state-of-arts papers in high-impact journals. In addition to the recommended papers, there are more papers about that, so find and add more.

Advanced Functional Materials 29.6 (2019):

Advanced Materials 32.15 (2020): 1905279

Round 2

Reviewer 1 Report

Please use arrows or any other symbols to identify/mark the fabric thickness measured on the representative SEM images.
